# The effect of biomass smoke exposure on quality-of-life among Ugandan patients treated for tuberculosis: A cross-sectional analysis

**Sophie Wennemann**[1]*, **Bbuye Mudarshiru**[2], **Stella Zawedde-Muyanja**[3], **Trishul Siddharthan**[4], **Peter D. Jackson**[5]

1 Virginia Commonwealth University School of Medicine, Richmond, Virginia, United States of America, 2 Makerere University Lung Institute, Kampala, Uganda, 3 Infectious Disease Institute, Makerere University College of Health Science, Kampala, Uganda, 4 Division of Pulmonary and Critical Care, University of Miami, Miami, Florida, United States of America, 5 Division of Pulmonary Critical Care, Virginia Commonwealth University, Richmond, Virginia, United States of America

* sophie.wennemann@gmail.com

**Data Availability Statement:** All data used as well as a key have been included in the submission as supporting information files.

## Abstract

More than half the global population burns biomass fuels for cooking and home heating, especially in low-middle income countries. This practice is a prominent source of indoor air pollution and has been linked to the development of a variety of cardiopulmonary diseases, including Tuberculosis (TB). The purpose of this cross-sectional study was to investigate the association between current biomass smoke exposure and self-reported quality of life scores in a cohort of previous TB patients in Uganda. We reviewed medical records from six TB clinics from 9/2019-9/2020 and conducted phone interviews to obtain information about biomass smoke exposure. A random sample of these patients were asked to complete three validated quality-of-life surveys including the St. Georges Respiratory Questionnaire (SGRQ), the EuroQol 5 Dimension 3 Level system (EQ-5D-3L) which includes the EuroQol Visual Analog Scale (EQ-VAS), and the Patient Health Questionnaire 9 (PHQ-9). The cohort was divided up into 3 levels based on years of smoke exposure–no-reported smoke exposure (0 years), light exposure (1–19 years), and heavy exposure (20+ years), and independent-samples-Kruskal-Wallis testing was performed with post-hoc pairwise comparison and the Bonferroni correction. The results of this testing indicated significant increases in survey scores for patients with current biomass exposure and a heavy smoke exposure history (20 + years) compared to no reported smoke exposure in the SGRQ activity scores (adj. p = 0.018) and EQ-5D-3L usual activity scores (adj. p = 0.002), indicating worse activity related symptoms. There was a decrease in EQ-VAS scores for heavy (adj. p = 0.007) and light (adj. p = 0.017) exposure groups compared to no reported exposure, indicating lower perceptions of overall health. These results may suggest worse outcomes or baseline health for TB patients exposed to biomass smoke at the time of treatment and recovery, however further research is needed to characterize the effect of indoor air pollution on TB treatment outcomes.

**Funding:** Funder Name: NIH, CTSA award No. UL1TR002649 from the National Center for Advancing Translational Sciences to support data collection (P.J.) Funder Name: CHEST Foundation Grant Number: Chest Foundation/ATS Research Grant in COVID and Diversity Grant Recipient: Dr Peter Durham Jackson Funder Name: American Thoracic Society Grant Number: Chest Foundation/ATS Research Grant in COVID and Diversity Grant Recipient: Dr Peter Durham Jackson Funder Name: School of Medicine, Virginia Commonwealth University Grant Number: DOIM Pilot Grant Grant Recipient: Dr Peter Durham Jackson The funders had no role in study design, data collection and analysis, decision to publish, or preparation of the manuscript.

**Competing interests:** The authors have declared that no competing interests exist.

## Introduction

Burning biomass fuels for cooking, home-heating, etc. is a practice utilized by more than half of the global population and a common source of indoor air pollution, primarily in low-middle income countries [1]. In Uganda in 2021, only 0.7% of the population relied primarily on clean fuels and technologies for cooking [2]. Long-term exposure to household air pollution like biomass fuels has been linked to various pulmonary diseases including chronic obstructive pulmonary disease (COPD), pneumonia, and tuberculosis (TB) [1], and the World Health Organization (WHO) attributed 3.2 million deaths globally to household air pollution in 2019 [3]. While the effect of biomass smoke exposure on the risk of TB diagnosis is well-established, few studies have focused on patient reported outcomes (PROMS) related to post-cure TB and biomass smoke exposure. In this cohort study of TB patients in Uganda, we assessed the association between biomass smoke exposure and self-reported health outcomes using validated quality of life measures.

## Methods

### Ethics statement

Regulatory approval for this cross-sectional study was obtained from the Makerere University School of Medicine Research and Ethics Committee of the College of Health Sciences (Ref: MakSOMREC 2021–54) and the Virginia Commonwealth University IRB (HM20020265). All subjects who completed questionnaires by phone provided verbal consent via phone witnessed by a member of our Ugandan research team in accordance with local regulations regarding human subject interviews, consent was documented on paper records and in the REDCap database. Phone interviews were conducted by Ugandan research assistants with significant experience in medical research. Interviews were conducted in English or Luganda, the two official languages of Uganda. All QOL questionnaires were translated to Luganda by a medical interpreter and approved by the Makerere University SOMREC. All consenting procedures were approved by aforementioned regulatory bodies. During data collection authors PJ (principal investigator) and BM (study coordinator) had access to identifying information to allow additional data collection in the event of mis-keyed, missing or misclassified data. Following study completion, no author had access to identified data and all data analysis was performed on deidentified data to prevent bias.

### Data collection

This analysis was imbedded in a previously published study which begun February of 2022, medical records were reviewed for patients receiving TB care between September 2019–2020 at three urban and three rural TB clinics in Uganda [4]. The rural clinics included Jinja Regional Referral Hospital, Mubende Regional Referral hospital and Kiboga General Hospital. The urban clinics included Kiruddu National Referral Hospital-Kampala district, Kisenyi Health Centre IV-Kampala district and Mulago National Referral Hospital-Kampala district. In an effort to obtain a representative sample, these clinics represented urban and rural centers as well as smaller referral hospitals (Kiboga, Kisenyi) and large national referral hospitals (Mulago, Kirrudu). Demographic, social, and medical information was collected via chart review for 1,624 subjects. Of these subjects, 1320 had phone numbers available and phone calls were attempted. To mitigate selection bias, individuals who did not answer were called three times on different days, and family contact numbers within the charts were also called to attempt to reach them. Of these individuals, 54% (710/1320) were ultimately contacted and consented for interviews (Fig 1). During these interviews, patients reported biomass exposure

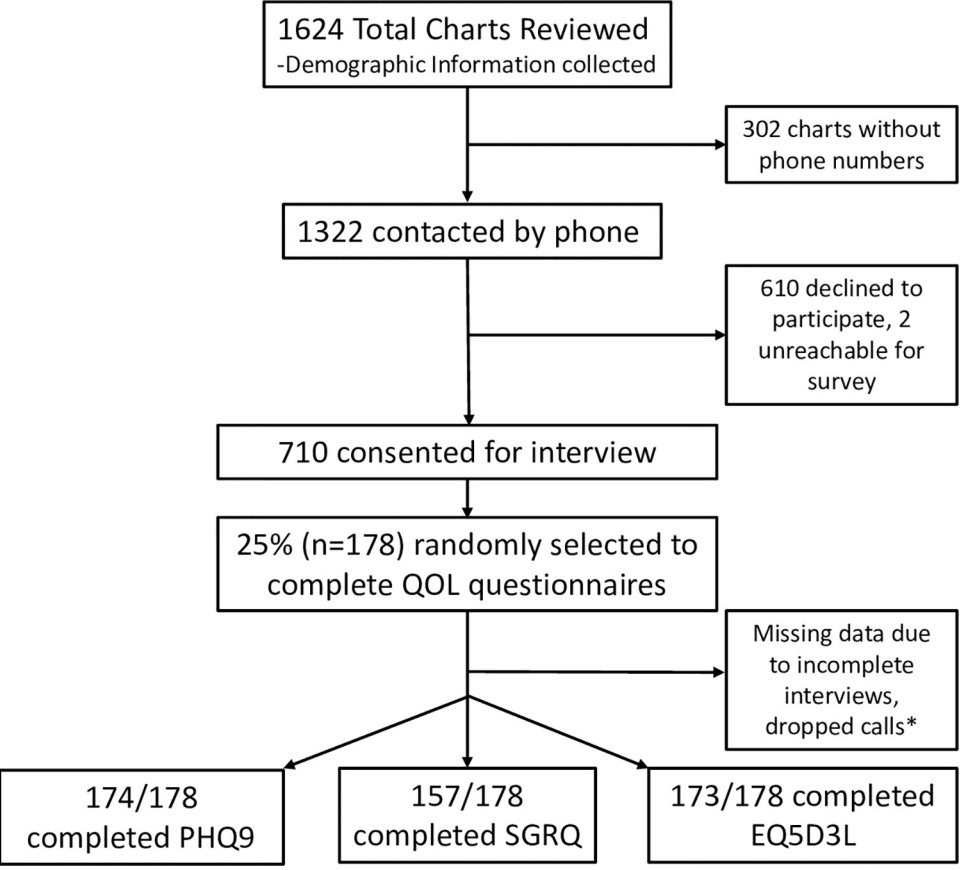

**Fig 1. Consort diagram summarizing data collection and exclusion.** SGRQ = St. George's Respiratory Questionnaire, EQ-5D-3L = EuroQol-5D-3L, EQ-VAS = EuroQol Visual Analog Scale, PHQ9 = Patient Health Questionnaire. *A small number of subjects were able to finish at least one survey, but not all three due to time constraints, dropped calls, etc. The SGRQ has a significantly lower number of responses than the PHQ9 and EQ-5D-3L because it requires significantly longer time to complete, and some subjects had issues with time constraints.

within their home including years used and past history of exposure. One quarter of patients (n = 178) were randomly selected to complete three validated quality of life (QOL) questionnaires during the phone interviews: the EuroQol-5D-3L (EQ-5D-3L), the St. George Respiratory Questionnaire (SGRQ), and the Patient Health Questionnaire (PHQ-9). Every fourth patient called and consented was selected to do these surveys, and this method was chosen to reduce the length of the interviews for the majority of the subjects within the study. Although a priori sample calculations were not performed, a conservative estimate based on data from Katoto et al. suggests that a sample of 146 would be sufficient to detect an OR of 2.7 assuming with power of 80% and type 1 error of 0.05, to detect worse symptoms in subjects previously treated for TB with biomass exposure [5].

The EQ-5D-3L measures 5 dimensions of health (mobility, self-care, usual activities, pain/discomfort, and anxiety/depression) using 3 levels (no, some, or extreme problems), with higher scores indicating more problems [6]. A visual analog scale (EQVAS) is also used, which has been validated to be delivered via phone [7] in which patients are asked to verbally quantify their current health on a scale of 0–100, with 0 indicating "the worst health you can imagine" and 100 indicating "the best health you can imagine." The SGRQ is a 16-question survey that calculates 3 weighted component scores including symptoms (frequency and severity), activity

(activities limited by respiratory symptoms), and impacts (social/psychological disturbances of airway disease), as well as a total score. Higher scores indicate increased severity and/or frequency of symptoms, greater limitations on activities, and more severe social/psychological impacts, respectively [8]. The PHQ-9 is a 9-question survey used to quantify depression severity, with higher scores indicating increased severity [9]. Given the various facets of health that may be affected by biomass exposure and TB disease, we elected to use the EQ-5D-3L to measure overall health related quality of life; the St. George Respiratory Questionnaire (SGRQ), to specifically measure respiratory quality of life; the PHQ-9 to measure the mental health effects of TB infection. All of these questionnaires have been used in TB research previously and are validated within Uganda and used extensively in research in low-middle income countries [10–15].

## Statistical analysis

Patients with complete questionnaires were stratified into three groups based on their level of biomass smoke exposure, with 0 years considered "no reported exposure", 1–19 years considered "light exposure", and 20+ years considered "heavy exposure" (Table 1). The phrase "no reported" smoke exposure is used to describe patients who did not report burning biomass fuels indoors. This term was chosen to recognize passive exposures throughout their lifetime i.e., from public or community exposures. Twenty years was selected as the cutoff for heavy exposure as this correlated with the top quintile of our cohort. Subjects who did not complete any of the surveys were not included in the analysis. Separate analyses were run for current biomass exposure years and past biomass exposure years. Scores from each questionnaire were compared between groups using an independent-samples-Kruskal-Wallis test with post hoc pairwise comparison and the Bonferroni correction to counteract multiple comparisons.

**Table 1. Characteristics of all participants by exposure group.**

| Chart review: 1624 Charts total | | | | | |
|---|---|---|---|---|---|
| Age in years, median (IQR): 34 (26–43) | | | | | |
| Gender: Female: 37% (601/1624), Male 63% (1024/1623) | | | | | |
| Completed interviews with at least 1 complete survey: 174 | | | | | |
| **Variables** | **Overall** | **No reported exposure (n = 106)** | **Light exposure (1–19 years) (n = 31)** | **Heavy exposure (20+ years) (n = 37)** | **P-values** |
| Age in years, median (IQR): | 36 (IQR 26–47) | 30 (37–23) | 30 (25–40) | 41 (32–50) | <0.001* |
| **Gender** | | | | | |
| Female | 62 (35.6%) | 29 (27.4%) | 17 (54.8%) | 16 (43.2%) | 0.011** |
| Male | 112 (64.4%) | 77 (72.6%) | 14 (45.2%) | 21 (56.8%) | |
| **Quantiles of SES** | | | | | |
| Lower | 66 (37.9%) | 45 (42.5%) | 13 (41.9%) | 8 (21.6%) | 0.092** |
| Middle | 60 (34.5%) | 38 (35.8%) | 8 (25.8%) | 14 (37.8%) | |
| Upper | 48 (27.6%) | 23 (21.7%) | 10 (32.3%) | 15 (40.5%) | |
| **Smoking history (cigarettes, cigars, or pipes)** | | | | | |
| No | 140 (80.5%) | 86 (81.1%) | 24 (77.4%) | 30 (81.1%) | 0.895** |
| Yes | 34 (19.5%) | 20 (18.9%) | 7 (22.6%) | 7 (18.9%) | |

SES: socioeconomic status, IQR: Interquartile Range

*Kruskal-Wallis test used

**Chi-square test used

We tested for confounding by age, income, and history of smoking cigarettes, pipes, or cigars (yes/no) using multivariate linear regression analysis with p <0.05 considered significant. These confounders were chosen for analysis as they are known to have independent effects on quality of life related to health, respiratory health and depression. Age and tobacco use are directly linked to changes in pulmonary physiology and function, which could have affected overall differences in results between groups if not assessed. Income is a common confounder for biomass exposure and QOL data in low-middle income countries. The medical charts at the TB clinics were also reviewed and queried for cardiopulmonary comorbidities that could potentially confound results, including heart failure, other heart diseases, asthma, chronic bronchitis, emphysema, and COPD. Medical history and presence of these comorbidities were also confirmed directly with the patient via phone interview. Of the subjects who had completed surveys and were included in the analysis, two had diagnosed asthma and two had diagnosed heart failure, with no other relevant comorbidities. Due to the lack of significant difference in pre-existing cardiopulmonary disease between biomass smoke groups, confounding analysis was not performed for these comorbidities.

## Results

For all questionnaires, there was no significant difference between exposure groups based on past biomass smoke exposure. In all the analyses outlined below, patients were grouped based on current biomass smoke exposure years. The SGRQ was completed by 157 patients of whom the median age was 33 (interquartile range IQR 25.5–41.5) and 66.2% were male. Of these patients, 33 had heavy biomass smoke exposure (21%), 29 had light exposure (18.5%), and 95 had no reported exposure (60.5%). The total scores produced differences between groups on the initial Kruskal-Wallis test (p = 0.047) but not the adjusted pairwise comparison (SE = 9.096, adj. p = 0.057) (Table 2). There were differences in the isolated SGRQ activity scores between the heavy exposure versus no reported exposure groups (SE = 7.869 adj. p = 0.018) (Table 2, Fig 2A). Linear regression indicated age (SE = 0.155, p = 0.951), income (SE = 6.34E-6, p = 0.972), and smoking history (standard error SE = 0.176 p = 0.155) were not significant confounders. There were no significant differences between groups for isolated SGRQ impact scores (p = 0.071) or symptoms scores (p = 0.191) (Table 2).

The EQ-5D-3L was completed by 173 patients of whom the median age was 33 (IQR 26–42) and 64.7% were male. Of these patients, 37 had heavy biomass smoke exposure (21.4%), 31 had light exposure (17.9%), and 105 had no reported exposure (60.7%). There were increases in the EQ1 mobility domain scores between groups (p = 0.001), but linear regression showed age (SE = 0.001, p = 0.004) and biomass exposure (SE = 0.009, p = 0.004) both had significant contributions to these differences. There were increases in usual activity domain scores (EQ3) between heavy exposure and both no reported exposure (SE = 6.208, adj. p = 0.002) and light exposure (SE = 7.906, adj. p = 0.041) groups (Table 2, Fig 2B). Linear regression showed no confounding by age (SE = 0.003, p = 0.102), income (SE = 1.265 E-7, p = 0.252) or smoking history (SE = 0.077, p = 0.110). The remaining EQ-5D-3L domains showed no significant differences.

There was a decrease in EQ-VAS scores in the light (SE = 9.123, p = 0.017) and heavy (SE = 8.533, p = 0.007) exposure groups compared to the no-reported exposure group (Table 2, Fig 3). Age (SE = 0.117, p = 0.195), income (SE = 4.9E-6, p = 0.744), and smoking history (SE = 2.943, p = 0.063) were not significant confounders.

The PHQ-9 was completed by 174 patients, of whom the median age was 36 (IQR 26–46.75) and 13.2% were male. Of these patients, 37 had heavy biomass smoke exposure (21.3%), 31 had light exposure (17.8%), and 106 had only no-reported smoke exposure (60.9%). There

**Table 2. Analyses of differences in survey scores between exposure groups.**

| Survey | N | Kruskal-Wallis Test | | Pairwise comparison* | | | | | |
| | | Test Statistic | Asymptotic Sig. (2-sided-test) | Comparison groups | Test Statistic | Std. Error | Std. Test Statistic | Sig. | Adj. Sig.** |
|---|---|---|---|---|---|---|---|---|---|
| SGRQ Activity score | 157 | 8.357 | 0.015 | No reported-Light exposure | -12.785 | 8.262 | -1.547 | 0.122 | 0.365 |
| | | | | No reported-Heavy exposure | -21.617 | 7.869 | -2.747 | 0.006 | 0.018 |
| | | | | Heavy exposure-Light exposure | -8.833 | 9.913 | -0.891 | 0.373 | 1 |
| SGRQ Total score | 157 | 6.118 | 0.047 | No reported-Light exposure | -12.756 | 9.551 | -1.336 | 0.182 | 0.545 |
| | | | | No reported-Heavy exposure | -21.342 | 9.096 | -2.346 | 0.019 | 0.057 |
| | | | | Heavy exposure-Light exposure | -8.586 | 11.458 | -0.749 | 0.454 | 1 |
| EQ5D3L Activity domain | 173 | 12.022 | 0.002 | No reported-Light exposure | -1.714 | 6.637 | -0.258 | 0.796 | 1 |
| | | | | No reported-Heavy exposure | -21.208 | 6.208 | -3.416 | <0.001 | 0.002 |
| | | | | Heavy exposure-Light exposure | -19.494 | 7.906 | -2.466 | 0.014 | 0.041 |
| EQ-VAS score | 173 | 14.355 | <0.001 | No reported-Light exposure | 2.957 | 12.069 | 0.245 | 0.806 | 1 |
| | | | | No reported-Heavy exposure | 30.521 | 9.477 | 3.221 | 0.001 | 0.004 |
| | | | | Heavy exposure-Light exposure | 27.563 | 10.132 | 2.72 | 0.007 | 0.02 |
| PHQ9 score | 174 | 4.923 | 0.085 | - | - | - | - | - | - |

SGRQ = St. George's Respiratory Questionnaire, EQ-5D-3L = EuroQol-5D-3L, EQ-VAS = EuroQol Visual Analog Scale, PHQ9 = Patient Health Questionnaire

*Each row tests the null hypothesis that the Sample 1 and Sample 2 distributions are the same.

Asymptotic significances (2-sided tests) are displayed. The significance level is 0.050.

**Significance values have been adjusted by the Bonferroni correction for multiple tests.

was no significant difference in mean PHQ-9 scores between biomass smoke exposure groups in the initial test (p = 0.085).

## Discussion

This cross-sectional study compared self-reported QOL scores in a cohort of Ugandan patients with varying degrees of biomass smoke exposure previously treated for TB. Our analysis centered specifically on the interplay of QOL after TB infection in those with biomass smoke exposure due to the growing interest in post-TB lung disease, which is defined as respiratory symptoms and pulmonary dysfunction after microbiologic cure. Our results show that previous TB patients with current heavy biomass smoke exposure in this sample reported worse symptoms related to activity on both the SGRQ and EQ-5D-3L, and those with any level of current biomass smoke exposure reported worse overall health than those with no-reported exposure on the EQVAS.

Pulmonary disease from biomass smoke is likely due to airway inflammation mediated by reactive oxygen species and proinflammatory cytokines. Similar to cigarette smoke, biomass smoke induces increased expression of matrix metalloproteases linked to obstructive lung disease [16–18]. Previous studies have shown structural lung defects (i.e., reduced pulmonary

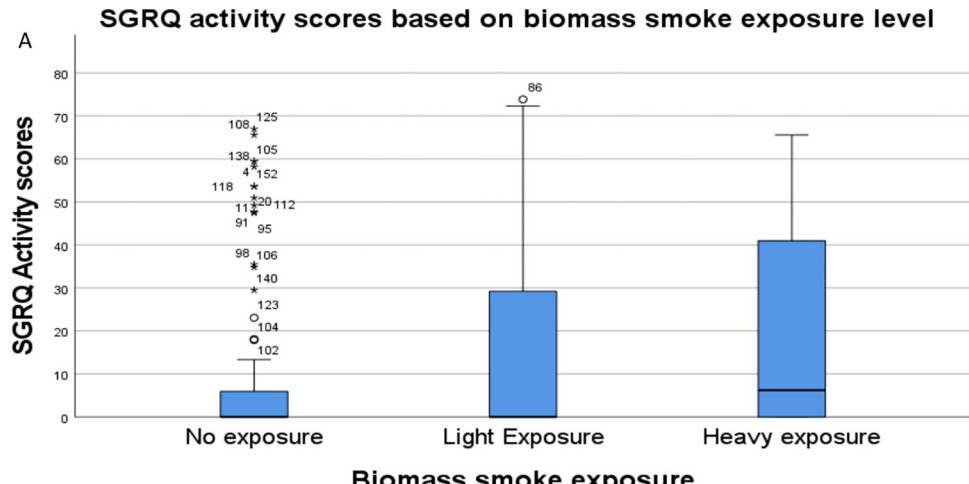

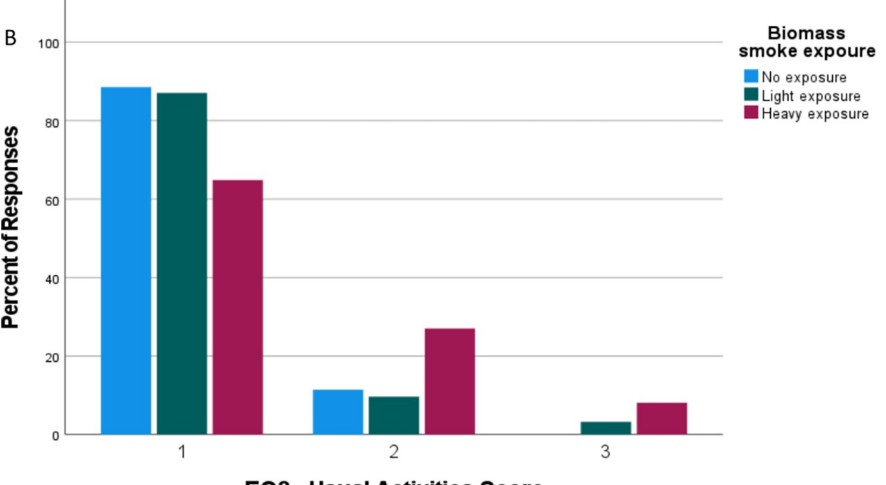

**Fig 2.** **(A)** St. George Respiratory Questionnaire (SGRQ) activity scores in patients with heavy, light, and no-reported biomass smoke exposure. The heavy exposure group had significantly higher scores than the no reported exposure group (SE = 7.869 adj. p = 0.018), indicating greater activity limitations. **(B)** Distribution of scores on the EuroQol-SD-3L Usual Activities questionnaire between heavy, light, and no reported exposure groups. The heavy exposure group reported significantly higher scores than both the no reported exposure (SE = 6.208, adj. p = 0.002) and light exposure (SE = 7.906, adi. p = 0.041) groups, indicating more activity limitations.

small vessel area, small airway remodeling) on CT scans of healthy individuals with biomass smoke exposure [19]. Increased prevalence of pulmonary symptoms and airway inflammation have also been found in patients with long-term biomass smoke exposure [20]. These effects of biomass smoke increase the pre-test probability of lung disease within this population and thus may affect TB outcomes, however, alternate explanations including impaired immunity from biomass smoke exposure may also contribute to QOL following TB treatment [21].

There are very few published studies investigating quality of life in Ugandan TB patients after cure. To our knowledge, this is one of the first studies investigating this in the specific context of biomass smoke exposure. In a cross-sectional study of obstructive lung disease and QOL after cure of multi-drug-resistant TB at one rural and one urban hospital in Uganda, Nuwagira et al., found that patients reported high rates of COPD and poor mental and physical

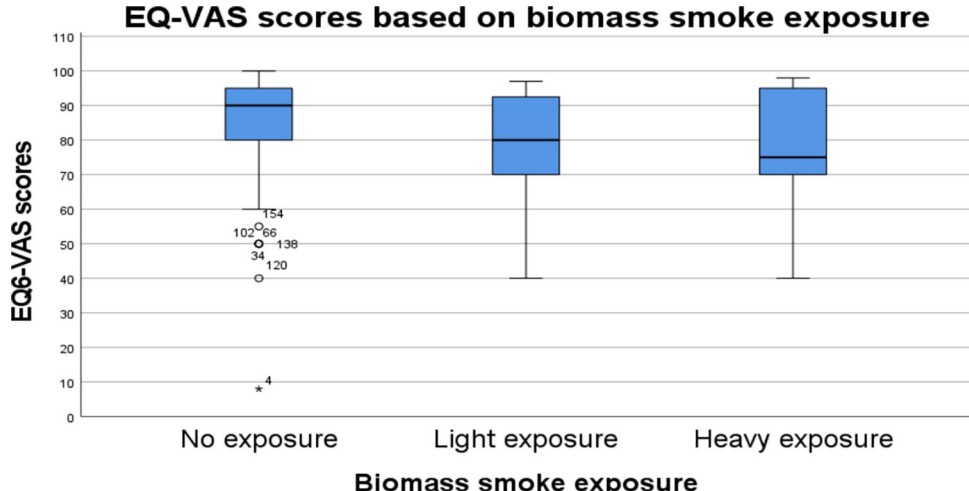

**Fig 3. EuroQol Visual Analogue Scale (EQ6-VAS) scores in patients with heavy, light. and no-reported biomass smoke exposure.** Patients in heavy (SE = 8.533, p = 0.007) and light (SE = 9.123, p = 0.017) exposure groups had significantly lower scores compared to the no exposure group, indicating worse perception of their overall health.

health as assessed by the Medical Outcomes Survey for HIV (MOS-HIV). They also found the median total score on the St. George's Respiratory Questionnaire (SGRQ) to be normal, but did not report data about the individual domains of the SGRQ [22]. It is important to note that this study reported median QOL scores in a standalone fashion without comparison to another group, whereas our study used these scores as a relative measure to compare QOL between groups with different risk profiles. However, the overall results of this study highlight the possible lasting impact of TB on perceived quality of life, even post-cure. A similar finding was reported by Daniels et al. in their study of quality of life in post-cure TB patients in Breede Valley District, South Africa [23].

Given the cross-sectional nature of this study, we are unable to determine if differences in survey results represent pre-existing poor health related to biomass smoke or worse treatment outcomes in previous TB patients. Key limitations of this study include the lack of a control group without previous TB with varying rates of biomass exposure to determine the magnitude of our findings within previous TB patients specifically. Given the lack of baseline spirometric data for this cohort, we cannot determine their pulmonary health prior to TB to exclude baseline lung disease as a confounder. While this is a significant limitation, the logistical challenges of obtaining pulmonary function testing prior to data collection in a random sample of tuberculosis patients is significant and beyond the scope of this study. Of note, subjects with a reported history of exposure did not have significant differences in their QOL surveys indicating that current exposure and not past history and baseline lung disease may be contributing to this finding.

Selection of our sample raises several limitations in that only 53.7% of charts with phone numbers were able to be contacted and consented to interview; additionally, within the 178 randomly selected to complete QOL surveys, 12% (21 subjects) were unable to successfully complete all three surveys due to time constraints, dropped calls, etc. As a result, the SGRQ ended up having a much lower completion rate than the EQ-5D-3L and the PHQ9 due to its length. These discrepancies could represent selection bias if systematic differences (i.e., lower SES, more demanding employment, comorbid conditions) in the characteristics of individuals who were eligible and participated in the study differed from those who were eligible but did not participate in the study. Collecting surveys during in-person meetings or appointments

may have mitigated this issue, however in-person collection was limited in our study period due to COVID-19 restrictions. Other possible sources of bias include recall bias both while completing the questionnaires and when asked about numbers of years exposed to biomass smoke. This limitation is inherent to such work however, and collecting longitudinal personal information on objective biomass smoke exposure in a cohort of subjects who will go on to have TB is logistically challenging. Finally, other factors such as ambient air pollution, stove type, stove age, and ventilation factors, which provide great variation in air pollutant concentrations and therefore degree of exposure [24], were not included in our analysis. While basic details about these factors were collected, the number of subjects within each group when separated by fuel type, ventilation, and stove type were too small to have adequate power for analysis.

Nevertheless, it is still plausible to consider how biomass smoke exposure could worsen post TB-cure lung impairment and contribute to worse outcomes in previous TB patients. Up to half of cured TB patients already experience ongoing symptoms and have abnormal spirometric tests post-cure [18], which could reasonably affect their ability to perform daily life tasks that require more physical exertion. Given the proposed mechanism of biomass smoke related pulmonary disease, it is possible that greater exposure to biomass smoke could act as potential risk factor for ongoing pulmonary symptoms and suboptimal treatment outcomes for TB. While this study is not definitive, it provides insight into the potential contribution of biomass smoke exposure as a risk factor for worse QOL after TB treatment. Additional research may be warranted to further investigate the interplay between biomass smoke exposure and post-TB lung disease.

## Supporting information

**S1 Checklist. STROBE statement—checklist of items that should be included in reports of observational studies.**
(DOCX)

**S1 Data. De-identified data.**
(XLSX)

**S1 Datakey. Data key.**
(PDF)

## Author Contributions

**Conceptualization:** Trishul Siddharthan, Peter D. Jackson.

**Data curation:** Bbuye Mudarshiru.

**Formal analysis:** Sophie Wennemann.

**Funding acquisition:** Peter D. Jackson.

**Investigation:** Sophie Wennemann, Bbuye Mudarshiru, Stella Zawedde-Muyanja, Trishul Siddharthan, Peter D. Jackson.

**Methodology:** Sophie Wennemann, Stella Zawedde-Muyanja, Trishul Siddharthan.

**Project administration:** Bbuye Mudarshiru, Stella Zawedde-Muyanja.

**Supervision:** Bbuye Mudarshiru, Stella Zawedde-Muyanja, Trishul Siddharthan, Peter D. Jackson.

**Validation:** Trishul Siddharthan.

**Writing – original draft:** Sophie Wennemann.

**Writing – review & editing:** Trishul Siddharthan, Peter D. Jackson.

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
