## [Decision Letter · Decision Letter 0]

27 Sep 2023

PGPH-D-23-01407

Research Letter: The Effect of Biomass Smoke Exposure on Quality-Of-Life among Ugandan Patients treated for Tuberculosis

Dear Dr. Sophie Wennemann,

Thank you for submitting your manuscript to PLOS Global Public Health. After careful consideration, we feel that it has merit but does not fully meet PLOS Global Public Health’s publication criteria as it currently stands. Therefore, we invite you to submit a revised version of the manuscript that addresses the points raised during the 

We look forward to receiving your revised manuscript.

Kind regards,

Reginald Quansah, Ph.D.

Academic Editor

Journal Requirements:

1.In the ethics statement in the Methods, you have specified that verbal consent was obtained. Please provide additional details regarding how this consent was documented and witnessed, and state whether this was approved by the IRB"

2. We have noticed that you have uploaded Supporting Information files, but you have not included a list of legends. Please add a full list of legends for your Supporting Information files after the references list.

**Additional Editor Comments (if provided)**:

Please, respond to all the reviewers' comments particularly, reviewer#3. In addition to reviewer#3 comments on exposure definition, note that the way in which exposure is defined will introduce serious exposure misclassification. (see Maggie L. Clark et al 2010:Indoor air pollution, cookstove quality, and housing characteristics in two Honduran communities. Environmental Research 110 (2010) 12–18) and needs to be discussed

Reviewers' comments:

Reviewer's Responses to Questions

**Comments to the Author**

1. Does this manuscript meet PLOS Global Public Health’s publication criteria? Is the manuscript technically sound, and do the data support the conclusions? The manuscript must describe methodologically and ethically rigorous research with conclusions that are appropriately drawn based on the data presented.

Reviewer #1: Partly

Reviewer #2: Yes

Reviewer #3: Yes

Reviewer #4: Yes

2. Has the statistical analysis been performed appropriately and rigorously?

Reviewer #1: Yes

Reviewer #2: No

Reviewer #3: Yes

Reviewer #4: Yes

3. Have the authors made all data underlying the findings in their manuscript fully available (please refer to the Data Availability Statement at the start of the manuscript PDF file)?

Reviewer #1: Yes

Reviewer #2: Yes

Reviewer #3: Yes

Reviewer #4: Yes

4. Is the manuscript presented in an intelligible fashion and written in standard English?

Reviewer #1: Yes

Reviewer #2: Yes

Reviewer #3: No

Reviewer #4: Yes

5. Review Comments to the Author

Reviewer #1: The article meets a recognized need of patient related data on quality of life of Uganda patients with TB exposure and treatment who might have concomitant exposure to indoor air pollution from biomass fuel.Unfortunately, in its current state it partially fails to meet the criteria for PLOS Global public health journal publication and has a flawed manuscript with conclusions that are supported by a seemingly compromised data collection process. The cross sectional study design for the primary research question posed projects an ethically and methodologically sound design with limitations and weaknesses that can be fully analyzed and presented. However, In my opinion, data collection process is a science which needs mastery execution to validate the data collected.

An attempt at statistical analysis of the data is seen through the mathematical formulas to infer or defer any data correlation or association but its meaning remain tainted with the aforementioned shortcomings. Additionally, a comment on data availability cannot be made at this point as the reviewer failed to access the supplementary data in its .xlx format.

In attempt to revise the aforementioned shortcomings, the following recommendations can be considered.

Line 32- the use of three data collection tools is commendable particularly in consideration of the type and quality of data that needs to be gathered to answer the posed question, however, a critical analysis of the data collection tools themselves needs to be provided in the form of a rationale for why they were selected and their advantages and disadvantages to the task they are being set to accomplish e.g how the tool depends on recall bias, language and understanding etc.

Line 63- author’s access to unblinded data is mentioned but is not acknowledged as a weakness in the limitations section. This mention as a weakness is validated by the potential author report bias that could affect the data collection and synthesis process. Additionally a detailed mentioning of which authors had access to the information would strengthen evidence based on a critical analysis of the data collectors skill/qualification to do so.

Line 72 and 92- there is paucity of information on the high attrition rate of 46% eg why 46% did not complete the interview and what was or could be done to reduce this high rate.

Line 74- provide an adequate description and rationale of the randomization process to select the 1/4 selected to complete questionnaires AND detail on the implementation of the phone interviewing eg call duration, language used, data collector’s qualifications etc because this helps judge the quality of evidence collected especially considering how a ‘visual’ analog scale was conducted through a phone interview(line80).

Line 95- provide rationale why the selected confounders were selected from a very wide range of confounders that could potentially misrepresent the meaning of the collected data. eg the recognized and mentioned limitations that lack of spirometric data, pre-existing pulmonary function etc could also confound the result’s interpretation. in the same vein, the ‘chart review’ process for data collection on cardiopulmonary comorbidities was no fully described and might need clarification on which chart was reviewed and if it was the best one to use for such relevant data seeing as this would affect interpretation of the results.

Line 104- There is an obvious discrepancy on the total patients who completed each questionnaire and the existence of this difference is not explained. This might challenge the comparability of the tools used for data collection (line 148) if not challenging the accuracy and validity of the actual result concluded for each tool.

Reviewer #2: Rationale

• It is not clear why study was done among TB patients. QoL due to biomass fuel can be poor irrespective of the TB status.

• Is the objective descriptive or analytical?

Methods

• The rationale for light/heavy exposure cut-offs not clear

• What was the objective for which sample size was calculated?

• What were the assumptions for sample size calculation?

• Was sample size adequate for analytical study?

Results

• The first para should describe QoL as per various scales. Objectives seem descriptive, but the results seem like an analytical study.

• Whether the biomass fuel exposure was current or in the past? With access to clean fuels, people might have changed their cooking fuel in the recent years.

• Describe the data as per various domains

• Too many models with no clear message. It is not clear if sample size is sufficient for such analysis.

Discussion

• If the objective was to see whether TB patients with biomass exposure had worse outcomes, a comparison group of non-TB patients should have been included in the study design. In the absence of control group, no interpretation can be made whether QoL is any different for TB patients.

• Discussion should focus on how the QoL compared with other TB studies in the same country/ similar settings.

Reviewer #3: 1. Concern about study design :

The study design is announced in the the discussion section (Lines 145, 157). Isn't it better to specify this at an earlier stage, to fix the reader's mind?

Suggested places : abstract, methodology and eventually in the title.

2. Concerns about samples and discrepancies between some sections

a. What population is studied ?

The authors say: « Beginning in February of 2022, Medical records were reviewed for patients receiving TB care from September, 2019 to September, 2020 at three urban and three rural TB clinics in Uganda ». lines 65 to 67

The first idea that emerge from this paragraph is that the studied group is TB patients (i.e. currently undergoing treatment). However, the dates seem to point towards previous patients. If Uganda applies the same treatment durations as we know (6 or 12 months), the cohorts from September 2019 to September 2020 are expected to complete their treatment before data collection in February 2022. Your discussion seems to support my point of view when you write "... previously treated for TB (L145-146)...". However, immediately afterwards, you refer again to patients with TB L146-148...". Isn't TB curable? So people who have been treated and declared cured from TB are no longer patients. In my opinion, these two concepts cannot be used for each other, as they refer to very different study populations.

Suggestions: it's important to make a choice and maintain consistency by talking about either "TB patients" or " previous patients" and not both.

b. sample representativeness

The authors state that subjects were selected from 6 structures (3 urban and 3 rural). However, the authors do not say anything about the number of medical centers that care for TB patients in Uganda, or about the selection technique used to select these 6 structures (whether random or not). At the same time, the title, objective and conclusion seem to suggest that the study results are applicable to Ugandan TB patients (or previous TB patients).

Questions: Is the sample of TB patients drawn from these 6 medical centers representative of Ugandan TB patients? The secondary questions to which we would like to find answers in the methodology are: Are the 6 selected facilities representative of the facilities that care for TB patients in Uganda? What weight do the patients cared for in these 6 facilities represent on a national scale? What is the spatial distribution of these 6 facilities across Uganda?

Suggestion: It may be necessary to provide more details on sampling or to adapt the title, objective and conclusion.

c. Sample size

How was the sample size estimated ? If exhaustive sample, please specify it.

Line 72 « Of these individuals, 54% (710/1320) and 672 completed interviews ». Is this sentence complete ? Finally, how much completed interviews :710 or 672 ?

Suggestion : It may be useful to add a flow chart showing the steps used to select the subjects included in the study and the stratification of respondents into the 3 groups.

3. Exposure concerns

Curiosity: what exactly does "zero exposure" mean? We know that there are many places of exposure, and that people can be passively exposed, for example in transport, in public places, during bush fires, ..... I don't personally believe that there are people with zero exposure. But if a threshold value has been used, if certain exposures have been minimized, this should be clearly indicated.

Exposure to smoke was investigated by starting from the interview period and working backwards, but the backward period was not defined in the paper. How was recall bias controlled? Similarly, the way in which years are counted needs to be clarified in the methodology, given that there may be periods of non-exposure due to travel, for example.

4. Results presentation

Is it not important to present some descriptive and inferential statistics in the tables?

Reviewer #4: Wenneman et al, in their paper “The effect of Biomass Smoke Exposure on Quality-Of-Life among Ugandan Patients treated for Tuberculosis” aimed to investigate the association between biomass smoke exposure and self-reported quality of life scores in a cohort of TB patient in Uganda. For this study, participants had been taken from the sample of a cohort study in Uganda on the effect of biomass smoke exposure on the risk of TB diagnosis. Here, researchers reviewed medical records of patients who were taking TB care from September, 2019 to September, 2020 from three unban and three rural TB clinics in Uganda. With verbal consent from the participants over phone, researchers randomly selected participants to complete three validated quality-of-life surveys including the St. Georges Respiratory Questionnaire (SGRQ), the EuroQol 5 Dimension Level system (EQ-5D-3L) and Patient Health Questionnaire 9 (PHQ-9). They also compared the self-reported quality-of-life of the TB patients with the exposure to biomass smoke that categorized into three groups depending on years of smoke exposure – no exposure (0-years), light exposure (1-19 years), and heavy exposure (20+ years). In aiming to obtain the result, analyzers performed independent-samples-Kruskal-Wallis testing with post-hoc pairwise comparison and the Bonferroni correction. In this paper they revealed significant relationship of the quality-of-life of TB patients on treatment with exposure of biomass smoke. After critical analysis of the findings of the survey and bio-statistical analysis, they were able to discover the relationship of biomass smoke exposure and quality of life of TB patients.

The strength of this paper is that it addressed an interesting and contemporary question, and was able to identify the pessimistic impacts of biomass smoke on the quality of life of TB patients. This article also inspired us to think about the adverse consequences of other factors on health outcomes of TB patients even those are on anti-Tubercular medications. This study also clearly demonstrated its limitations and suggested for further evaluation through another similar studies.

However, this study includes some deficiencies. One of the weaknesses of this article is occasional incomprehensiveness which establishes unclear logical links between concepts. In addition to this, method of sampling seems to be little confusing, particularly in line no 72. Another possible weakness could be the discussion part which might be more descriptive and topic oriented.

In spite of having some weaknesses of this article, it is published worthy as it has added a new concept in the field of research.

6. PLOS authors have the option to publish the peer review history of their article (what does this mean?). If published, this will include your full peer review and any attached files.

**Do you want your identity to be public for this peer review?** For information about this choice, including consent withdrawal, please see our Privacy Policy.

Reviewer #1: No

Reviewer #2: No

Reviewer #3: No

Reviewer #4: No

---

## [Decision Letter · Decision Letter 1]

8 Jan 2024

PGPH-D-23-01407R1

The Effect of Biomass Smoke Exposure on Quality-Of-Life among Ugandan Patients treated for Tuberculosis: a cross-sectional analysis

Dear Dr. Sophie Wennemann,

Thank you for submitting your manuscript to PLOS Global Public Health. After careful consideration, we feel that it has merit but does not fully meet PLOS Global Public Health’s publication criteria as it currently stands. Therefore, we invite you to submit a revised version of the manuscript that addresses the points raised during the review process.

We look forward to receiving your revised manuscript.

Kind regards,

Reginald Quansah, Ph.D.

Academic Editor

Journal Requirements:

Additional Editor Comments (if provided):

Reviewers' comments:

Reviewer's Responses to Questions

**Comments to the Author**

1. If the authors have adequately addressed your comments raised in a previous round of review and you feel that this manuscript is now acceptable for publication, you may indicate that here to bypass the “Comments to the Author” section, enter your conflict of interest statement in the “Confidential to Editor” section, and submit your "Accept" recommendation.

Reviewer #1: All comments have been addressed

Reviewer #2: All comments have been addressed

Reviewer #3: All comments have been addressed

2. Does this manuscript meet PLOS Global Public Health’s publication criteria? Is the manuscript technically sound, and do the data support the conclusions? The manuscript must describe methodologically and ethically rigorous research with conclusions that are appropriately drawn based on the data presented.

Reviewer #1: Yes

Reviewer #2: Yes

Reviewer #3: Yes

3. Has the statistical analysis been performed appropriately and rigorously?

Reviewer #1: Yes

Reviewer #2: Yes

Reviewer #3: Yes

4. Have the authors made all data underlying the findings in their manuscript fully available (please refer to the Data Availability Statement at the start of the manuscript PDF file)?

Reviewer #1: Yes

Reviewer #2: Yes

Reviewer #3: Yes

5. Is the manuscript presented in an intelligible fashion and written in standard English?

Reviewer #1: Yes

Reviewer #2: Yes

Reviewer #3: Yes

6. Review Comments to the Author

Reviewer #1: A significant improvement from the original submission. The revision addressed proposed recommendations and remains with a few areas for more improvement as pointed out below.

Line 99- “ although no a priori sample calculations were performed “ possible grammatical error.

Line 111- grammatical error, The EQ-5q-3l

Line 140- Give the meaning of the acronym IQR at the bottom of the table like what is done for SES.

line 173- The table format is not showing the last column after standard error (on the reviewer’s screen it appears just cut off from the end of the page).

Line 166- The kruskal-wallis one way analysis of variance results are missing the number before the decimal point and your reader may not know this is automatically a zero so a recommendation would be to be uniform.

Reviewer #2: (No Response)

Reviewer #3: I congratulate the authors who addressed my comments. I am satisfied.

7. PLOS authors have the option to publish the peer review history of their article (what does this mean?). If published, this will include your full peer review and any attached files.

**Do you want your identity to be public for this peer review?** For information about this choice, including consent withdrawal, please see our Privacy Policy.

Reviewer #1: **Yes: **Benson Tarisai Gombe

Reviewer #2: No

Reviewer #3: **Yes: **Gabriel Kyomba Kalombe

---

## [Editor Report · Decision Letter 2]

18 Jan 2024

The Effect of Biomass Smoke Exposure on Quality-Of-Life among Ugandan Patients treated for Tuberculosis: a cross-sectional analysis

PGPH-D-23-01407R2

Dear Dr. Sophie Wennemann,

We are pleased to inform you that your manuscript 'The Effect of Biomass Smoke Exposure on Quality-Of-Life among Ugandan Patients treated for Tuberculosis: a cross-sectional analysis' has been provisionally accepted for publication in PLOS Global Public Health.

Best regards,

Reginald Quansah, Ph.D.

Academic Editor